# Senescent Fibroblasts Generate a CAF Phenotype through the Stat3 Pathway

**DOI:** 10.3390/genes13091579

**Published:** 2022-09-02

**Authors:** Hao Li, Lei Qiu, Qing Liu, Zelong Ma, Xiaoli Xie, Ying Luo, Xiaoming Wu

**Affiliations:** 1Laboratory of Molecular Genetics of Aging & Tumor, Medical School, Kunming University of Science and Technology, Chenggong Campus, 727 South Jingming Road, Kunming 650500, China; 2Guizhou Provincial Key Laboratory of Pathogenesis & Drug Development on Common Chronic Diseases, School of Basic Medicine, Guizhou Medical University, Guiyang 550025, China

**Keywords:** fibroblast senescence, Stat3 pathway, cancer-associated fibroblasts

## Abstract

Aging has been recently reported to promote lung cancer initiation and progression. Senescent fibroblasts gain a cancer-associated fibroblast (CAF) phenotype, and exert a powerful influence on cancer behavior, such as tumor cell growth and metastasis. However, mechanisms linking fibroblast senescence with CAF activation remain poorly understood. Our study shows that senescent fibroblasts displayed CAF properties, including the highly expressed CAF markers, α-SMA and Vimentin, and CAF-specific factors, CXCL12, FGF10, IL6 and COL1A1, which significantly increased collagen contractile activity and promoted the migration and invasion of lung cancer cells, H1299 and A549. We were further able to show that CAF characteristics in senescent fibroblasts could be regulated by the Stat3 pathway. Intracellular ROS accumulation activates the Stat3 pathway during senescence. Thus, our findings indicate that senescent fibroblasts mediate a CAF function with the Stat3 pathway. We further propose a novel Stat3 dependent targetable mechanism, which is instrumental in mediating the migration and invasion of lung cancer cells.

## 1. Introduction

Lung cancer is the leading cause of cancer mortality worldwide [1]. In recent years, aging has been recognized as influencing the initiation and progression of lung cancer [2]. Subsequently, senescent fibroblasts, which are generated due to adaptation to aging, are associated with lung cancer progression and have attracted scholarly attention. It has been demonstrated that senescent fibroblasts are capable of CAF [3]. Therefore, there is some urgency in studying whether senescent fibroblasts potentially promote lung cancer progression to gain a CAF phenotype.

CAF are one of the most abundant stromal components and play a critical role in shaping the malignant tumor microenvironment (TME) [4,5]. CAF isolated from carcinomas secrete a variety of soluble signaling factors (such as chemokines and cytokines), which enhance cancer growth, invasion, metastases and evading immune responses in a number of solid cancers [6,7,8,9,10]. The major source of CAF are normal fibroblasts and highly expressed positive molecular markers such as α-SMA, FAP, S100A4, Vimentin and platelet-derived growth factor receptor-β (PDGFR-β). Despite their tumor-promoting properties, CAF remain poorly defined due to the heterogeneity of fibroblasts.

It has been shown that senescent stromal cells in the TME promote tumor aggressiveness and are a primary cause of therapy resistance [11,12,13,14]. Accumulating evidence suggests that senescence fibroblasts, which are generated by various stimuli, promote tumorigenesis and tumor growth by expressing a pro-tumorigenic factor known as the senescence-associated secretory phenotype (SASP) [15,16,17,18]. Human fibroblasts treated with granulin render a CAF phenotype [19]. In addition, cells that fail to enter senescence following exposure to senescence-inducing stress robustly express SASP factors [20]. These observations indicate that the mechanisms governing CAF function in senescent fibroblasts are dependent upon senescence. There is significant overlap between the pro-tumorigenic factors expressed in CAF and senescent cells. Despite the profound impact that pro-tumorigenic SASP has on tumor cell growth and progression, the mechanisms linking CAF activation with fibroblast senescence are not known.

In our previous study, we revealed the crucial role of Stat3-mediated signaling in CAF activation using mouse embryo fibroblasts from the *p53^S/S^* (p53^N236S^ mutation) mouse model [21]. Nevertheless, it is unknown whether the Stat3 pathway in senescent fibroblasts could play the same role. In this study, we analyzed fibroblasts cultured ex vivo, using mouse embryonic fibroblasts (MEFs) induced to senesce through an extended culture or DNA damaging stimuli. We found that induction of fibroblast senescence gains CAF properties, which promote the migration and invasion of lung cancer cell lines H1299 and A549. We also found that the repressed Stat3 pathway could rescue the senescent fibroblast-mediated biological features of CAF. We were further able to show that the Stat3 pathway was mediated through intracellular ROS accumulation and prolonged Stat3 phosphorylation. These results suggest that the regulatory mechanisms elucidated in CAF are applicable in senescent stroma [21]. Fibroblast senescence and lung cancer progression are closely linked processes, which impact on patient survival. Thus, we further propose a novel Stat3 dependent targetable mechanism that is instrumental in mediating the migration and invasion of lung cancer cells.

## 2. Materials and Methods

### 2.1. Mice

The WT mice involved in our study were in line with the Guide for the Care and Use of Laboratory Animals. Animal experiments and protocols were approved by the Animal Care and Use Committee of Kunming University of Science and Technology (approval number: PZWHK2019-0005).

### 2.2. Cell Lines and Treatments

As previously described [22], MEFs were harvested and prepared from individual day 13.5 embryos obtained from wild-type C57/B6 mice. Briefly, embryos were rinsed several times with sterile phosphate buffered saline (PBS), minced with a scalpel and digested with trypsin (0.25%) for 10 min at 37 °C, following removal of the head and liver. The trypsin was inactivated with the addition of DMEM (Gibco) and supplemented with 10% fetal bovine serum (FBS) (Hyclone, Logan, UT, USA). The embryo was plated into one 10 cm dish. The cells were passaged at a ratio of 1:3, and they would be senescent until reaching passage 7. The human NSCLC cell lines, H1299 and A549, were cultured in DMEM with 10% FBS.

Fibroblasts were treated with 5 µg/mL doxorubicin(dox) (Sigma, St. Louis, MO, USA) for 12 h. Cells were treated with 7.5 μM Stattic (Selleckchem, Houston, TX, USA) for 6 h. All cells were cultured at 37 °C with 5% CO_2_ and 3% O_2_. Routine tests for cell mycoplasma infection were performed and confirmed to be negative.

### 2.3. Transient siRNA Transfection

Fibroblasts were transfected with siRNA (three siStat3 sequences, SiStat3-1:5′-CUGGAUAACUUCAUUAGCA-3′; SiStat3-2:5′-UAUCAUCGACCUUGUGAAA-3′; SiStat3-3:5′-CCAACGACCUGCAGCAAUA-3′). Random sequences were used as a negative control (NC). Briefly, fibroblasts were seeded in a 10 cm dish, grown to 70–80% confluence and then transfected with 20 nM siRNA using lipofectamine 3000 (Invitrogen, Carlsbad, CA, USA). Then, 48 h later, cells were collected for Western blot analyses to determine knockdown efficiency.

### 2.4. Quantitative Reverse Transcription–Polymerase Chain Reaction (qRT-PCR)

Total RNA was isolated from fibroblasts using Trizol (Invitrogen) according to the manufacturer’s protocol. RNA quantity was determined by agarose gel electrophoresis and spectrophotometry. Single-stranded complementary DNA (cDNA) was obtained from a reverse transcription of 600 ng of RNA using an RT-PCR kit (BD Biosciences, Franklin, NJ, USA). A qRT-PCR was performed using SYBR (Takara Bio, Osaka Prefecture, Osaka City, Japan) according to the manufacturer’s instructions. Data were normalized to β-actin expression. Each sample was analyzed in triplicate. Gene expression was analyzed using the 2^−ΔΔCt^ method.

The sequences of each forward (F) and reverse (R) primer used for qRT-PCR were as follows:

CXCL12-F: 5′-TGCATCAGTGACGGTAAACCA-3′,

CXCL12-R: 5′-CACAGTTTGGAGTGTTGAGGAT-3′;

IL6-F: 5′-TAGTCCTTCCTACCCCAATTTCC-3′,

IL6-R: 5′-TTGGTCCTTAGCCACTCCTTC-3′;

FGF10-F: 5′-CACATTGTGCCTCAGCCTTTC-3′,

FGF10-R: 5′-CTCTATTCTCTCTTTCAGCTTAC-3′;

Col1A1-F: 5′-CATTGTGTATGCAGCTGACTTC-3′,

Col1A1-R: 5′-CGCAAAGAGTCTACATGTCTAGG-3′;

β-actin-F: 5′AGAGGGAAATCGTGCGTGAC-3′,

β-actin-R: 5′-CAATAGTGATG ACCTGGCCGT-3′;

### 2.5. Western Blotting

Fibroblasts were lysed in an RIPA buffer containing Protease Inhibitor Cocktail (Roche). Sample proteins (20 μg) were separated by SDS-PAGE and then transferred to PVDF membranes. After blocking in 10% nonfat milk for 1 h at room temperature, membranes were incubated with primary antibodies overnight at 4 °C or 2 h at room temperature. The membranes were then incubated with horseradish peroxidase-labeled secondary antibodies and visualized with ECL. The following primary antibodies were used: anti-p-Stat3(1:1000, CST, Danvers, MA, USA), anti-Stat3(1:1000, CST, Danvers, MA, USA), anti-α-SMA(1:4000, Abcam, Cambridge, UK), anti-p53(1:800, Santa, CA, USA), anti-vimentin(1:1000, Abclonal, Wuhan, China), anti-p21(1:1000, BD, Franklin, NJ, USA), anti-p16(1:1000, SAB, College Park, MD, USA) and anti-GAPDH (1:3000, Abclonal, Wuhan, China). The relative expression was normalized to GAPDH expression.

### 2.6. Preparation of Conditional Media

Fibroblasts were seeded in DMEM supplemented with 10% FBS on a 10 cm^2^ dish and allowed to reach confluence. The medium was then replaced with serum-reduced DMEM with 0.1% FBS. Fibroblasts (WT p3, WT p7) were incubated at 37 °C for 24 h. The medium was then collected, centrifuged, filtered and stored at −80 °C for later use.

### 2.7. Migration and Invasion

Transwell chambers without Matrigel were used to examine the tumor cell ability of migration, and transwell chambers were pre-coated with 50 µL Matrigel to measure cell invasion (BD Biosciences, Franklin, NJ, USA). In brief, on the top of the inserts, tumor cells in a serum-free medium (1 × 10^4^ A549 or H1299) were added to 24-well transwell plates with 8 mm pores. On the bottom chamber, conditional media from fibroblasts (young fibroblasts: WT p3, senescent fibroblasts: WT p7) or WT p3, WT p7, WT p7 + NC (random sequences) and WT p7 + siStat3 with 10% FBS were added. After incubation for 18 h, the invaded cells were fixed with 40% methanol, stained with crystal violet, and 5 non-overlapping fields were imaged (Nikon Eclipse TI-E microscope, Tokyo, Japan) and quantified. Each experiment was repeated three times. Quantification of migration/invasion cells was calculated as the percentage of migration/invasion cells relative to the total number of tumor cells, which were seeded on the top of the inserts.

### 2.8. Fibroblast Contraction Assay

The contraction assay was performed to evaluate the contractility of fibroblast cells. Fibroblasts were embedded in 400 µL of type I collagen (Corning, Corning, NY, USA). The final cell density was 3 × 10^5^ cells/mL with 1 mg/mL collagen. This mixture was seeded in a 12-well plate, incubated for 24 or 48 h in a humidified incubator at 37 °C and imaged. To calculate the relative collagen contraction area (%), scanned images were quantified with Image J software.

### 2.9. Immunofluorescence Staining

Cells were seeded on glass coverslips, washed in PBS and fixed in 4% PFA (10 min, room temperature). Cells were permeabilized using Triton-X-100 (5 min, on ice), blocked with 2% BSA (1 h, room temperature) and incubated with primary antibodies (α-SMA, 1:500, Abcam) overnight at 4 °C. Cells were incubated with Alexa 568-coupled secondary antibody (1:1000, Jackson ImmunoResearch Laboratories Inc, Baltimore, Pennsylvania, USA) for 1 h at room temperature, counter-stained with DAPI (Beyotime, Shanghai, China) and mounted in the slide.

### 2.10. Reactive Oxygen Species Assay

Cells were incubated for 30 min with 10 μM dichlorofluorescein diacetate (DCFH-DA) in the dark and fixed using 4% paraformaldehyde. Flow cytometry analyses were performed using a FACS Calibur (BD Biosciences, Franklin, NJ, USA). Intracellular ROS caused oxidation of DCFH-DA, yielding the fluorescent product 2′,7′-dichlorofluorescein (DCF). Relative ROS levels were quantified as the mean florescence intensity of DCF. Fibroblasts treated with H_2_O_2_ were used as the positive control, and unstained fibroblasts were used as the negative control. All flow cytometry was analyzed using FlowJo software.

### 2.11. Senescence β-Galactosidase Staining

Senescence staining was measured with a Senescence Cells Histochemical Staining Kit (Sigma, St. Louis, MI, USA). Briefly, cells were fixed for 10 min at room temperature with 4% paraformaldehyde. After two washes in PBS, cells were incubated for 12 h at 37 °C in freshly prepared senescence-associated β-Galactosidase (SA-β-Gal) staining solution containing 1 mg/mL 5-bromo-4-chloro-3-indolyl b-D-galactopyranoside (X-Gal). Positively blue-stained senescent cells were counted.

### 2.12. Statistical Analysis

Student′s *t*-tests were used to assess the significance of differences between two groups of data (*p* < 0.05 is deemed significant).

## 3. Results

### 3.1. Senescent Fibroblasts Exert CAF Properties

Previous studies have demonstrated that fibroblast senescence is coupled with activated CAF properties [23,24,25]. To address this issue, we used the wild type (WT) MEF, induced to senesce using replication in culture. The late-passage (*p* = 7) MEF exhibited a classical morphology characterized by increased granularity and an enlarged and flattened shape. In addition, we confirmed senescence by the increase in fibroblasts, which were positively stained for senescence-associated β-gal (SA-β-gal, Figure 1A, 73.9 ± 12.3%). Consistently, a marked accumulation of senescence markers p53, p21 and p16 occurred in senescent fibroblasts compared to young fibroblasts (*p* = 3) (Figure 1B). In addition, signature CAF characteristics were acquired in these senescent fibroblasts. We observed that senescent fibroblasts significantly upregulated the expression of CAF markers, α-SMA and Vimentin (1B), and 97% senescent fibroblasts with α-SMA-positive staining (Figure 1A). Enhanced contractile activity is an important pathological feature of CAF; we examined whether senescent MEF could enhance the cell contractility of fibroblasts and cause mechanical stress similar to that of CAF. As expected, senescent MEF displayed markedly stronger contractile activity in a 3D collagen I matrix (Figure 1C). Collectively, we suggest that fibroblasts acquired CAF properties during senescence.

### 3.2. Senescent Fibroblasts Promote Lung Cancer Cell Migration and Invasion

Senescent fibroblasts may promote lung cancer progression. We then investigated whether senescent fibroblasts are able to influence lung cancer cell migration or invasion through paracrine signaling. H1299 and A549 cell migration and invasion were more affected by exposure to conditioned mediums (CM) from senescent fibroblasts than exposure to CM from young fibroblasts (Figure 2A,B), indicating the significant impact of senescent fibroblasts on tumor cell migration and invasion via secreted factors. These data are in line with our previous findings of a reciprocal interplay between CAF and lung cancer cells [21]. We previously discovered that the Stat3 pathway significantly contributes to CAF activation. Collectively, these data raise the possibility that the universal upregulation of the Stat3 pathway in fibroblasts may have a role in activating CAF characteristics.

### 3.3. Senescent Fibroblasts Exhibit CAF Properties through the Stat3 Pathway

We observed that many distinctive features of senescent fibroblasts resemble those of CAF [26]. Thus, we investigated whether in vitro senescent fibroblasts share the same regulatory pathway with CAF. To investigate this, we used WT MEF, induced to senesce using replicative senescence or treated with dox, to promote CAF phenotypes. Notably, we observed that similar to replicative senescence, fibroblasts induced to senesce by dox showed increased expression of CAF markers, α-SMA, and were contractile (Figure 3A,B). The senescence-inducing stimuli (replicative senescence, dox) similarly showed a highly increased expression of p-Stat3^Tyr705^ (Figure 2A). The Stat3 protein is a key factor in Stat3 pathway activation, and phospho-Stat3 (Tyr705) is the activated status of Stat3. Stat3 signaling can be inhibited using a specific inhibitor of Stat3 (Stattic), or the siRNA knockdown of Stat3 suppressed p-Stat3 upregulation in senescent fibroblasts (Figure 3A and Figure 4A). We have demonstrated that inhibition of the Stat3 pathway activity strongly compromised contractile activity in senescent fibroblasts (Figure 3B). Moreover, Stat3 pathway inhibition clearly abrogated senescent fibroblast-induced H1299 and A549 migration and invasion, suggesting that senescent fibroblasts render a CAF phenotype via the Stat3 pathway (Figure 4B). Accordingly, Stat3 was knocked down in senescent fibroblasts, resulting in a significant reduction in the level of Stat3 pathway-dependent CAF-specific factors, CXCL12, FGF10, IL6 and COL1A1 (Figure 4C). These data indicated that Stat3 pathway-dependent CAF-specific factors within the tumor microenvironment play a pivotal role in the migration and invasion of lung cancer cells in vitro.

### 3.4. Intracellular ROS Accumulation Activated Stat3 Phosphorylation in Senescent Fibroblasts

We next sought to elucidate the mechanism of Stat3 pathway activation in senescent fibroblasts. Previous studies have demonstrated that accumulated reactive oxygen species (ROS) activated Stat3 signaling, prompting us to hypothesize that Stat3 activation in senescent fibroblasts occurs through ROS generation [27]. Indeed, senescent fibroblasts, induced by a serial passage or dox, displayed increased reactive oxygen species (ROS) levels when compared to young fibroblasts (Figure 5A). Moreover, NAC reduced Stat3 phosphorylation by scavenging ROS (Figure 5B). From these experiments we concluded that high level ROS are involved in modulating CAF functions through the Stat3 pathway.

## 4. Discussion

Our understanding of the interaction between senescent fibroblasts and CAF has evolved in recent years. The increase in many aging-associated cancers has been linked with stromal fibroblast senescence and concomitant secreting diffusible factors, which are similarly induced in CAF [23]. While CAF show no signs of senescence and are characterized by increased cell proliferation [28,29], increasing evidence suggests that senescent fibroblasts are CAF in lung cancer [12,30]. Indeed, targeting senescence can decrease radioresistance in non-small cell lung cancer (NSCLC) [12]. However, the molecular mechanisms underlying CAF function activation in senescent fibroblasts remain largely unclear.

To account for the overlapping properties of senescent fibroblasts and CAF, we demonstrated that senescent fibroblasts play a role in lung cancer. Our results revealed that senescent fibroblasts promote lung cancer cell line migration and invasion. On the other hand, we confirmed that senescent fibroblasts enhance the contractility of fibroblasts. Together, these data identify senescent fibroblasts with an acquired CAF function. We have found evidence to indicate that senescent fibroblasts and lung cancer metastasis may be linked. However, we are not sure whether senescent fibroblasts have other pro-tumor functions. Further investigation is needed to verify this idea.

Next, we identified the mechanisms of senescent fibroblasts with acquired CAF properties. Our results show that senescent fibroblasts regulate CAF phenotypes, resulting in the augmented expression of α-SMA and Vimentin, which were regulated by the Stat3 pathway. Moreover, we used NAC to block ROS, finding that the Stat3 pathway was suppressed. This demonstrated that high level ROS rendered senescent fibroblast CAF properties through the Stat3 pathway. We are puzzled by the effect of ROS in Stat3 pathway activation. One possible mechanism might be that high level ROS enhanced activation of Stat3 phosphorylation by restraining its phosphatase SHP2, which is a negative regulator of Stat3. SHP2 is known to inhibit Stat3 phosphorylation; high levels of SHP2 and low levels of p-Stat3 in tumors correlate with longer cancer patient survival [31]. Further investigation is needed to verify this idea.

Of note, wt-p53 was activated in senescent fibroblasts (Figure 1B). There is increasing evidence that wt-p53 regulates the Stat3 pathway, although conflicting results have been reported. Whereas several studies have shown that wt-p53 increases Stat3 activation [32], other studies have reported that wt-p53 inhibits the Stat3 pathway [27,33]. However, whether wt-p53 in senescent fibroblasts plays a role in the regulation of the Stat3 pathway needs to be elucidated further.

In recent years, senescent fibroblasts/CAF have become a novel target in cancer treatment [12]. Novel strategies of combining CAF and senescent fibroblast targeting for cancer therapy should be of great interest. Identification of the Stat3 pathway as a pivotal participant in CAF function activation, both in normal fibroblasts and senescent fibroblasts, highlights that it could be a novel target for therapeutic interventions. However, a blockade of the Stat3 pathway may result in an impairment of tumor immune surveillance [34], and inhibition of the Stat3 pathway did not affect normal cell survival [35]. These different Stat3 pathway functions in tumorigenesis need to be taken into account in future clinical trials targeting the Stat3 pathway.

In conclusion, the findings of this study verify that senescent fibroblasts potentially promote lung cancer metastasis to gain a CAF phenotype, and provide evidence that intracellular ROS accumulation in senescent fibroblasts may activate the Stat3 pathway during CAF phenotype activation. Targeting senescent fibroblasts may be a novel and efficient lung cancer metastasis therapy.

## Figures and Tables

**Figure 1 genes-13-01579-f001:**
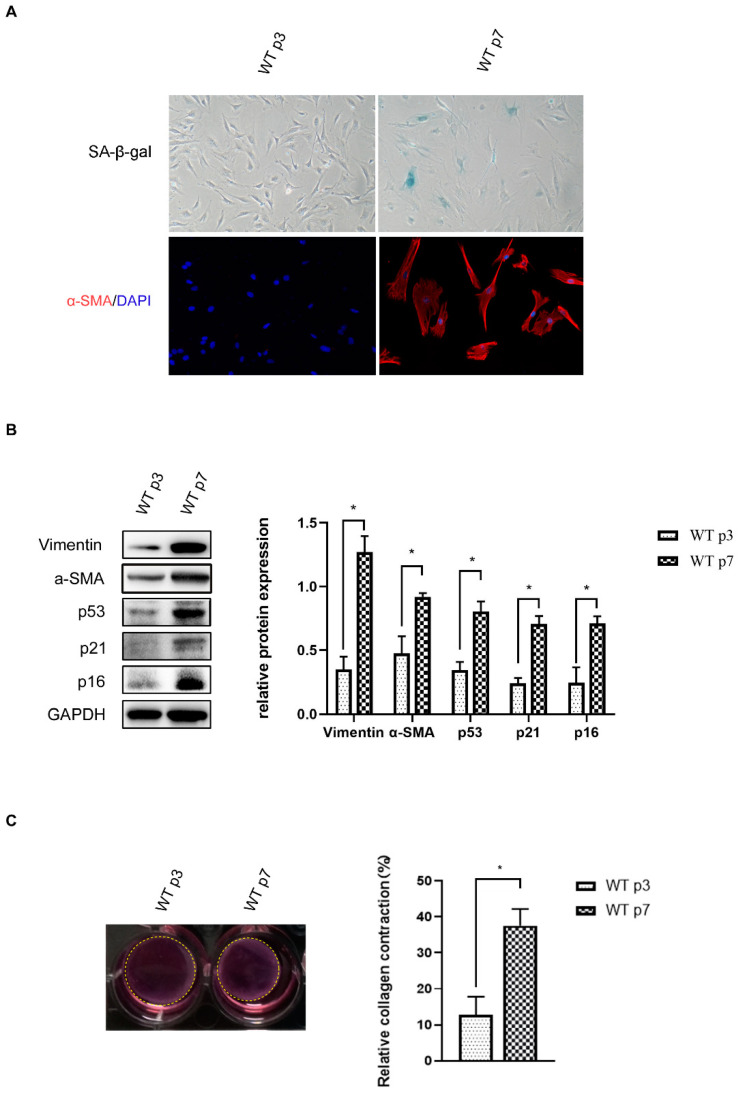
**Fibroblast****senescence coupled with acquisition of CAF properties.** (**A**). Representative images of SA-β-gal staining of early-passage fibroblasts (*p* = 3) and late-passage fibroblasts (p7). Late-passage fibroblasts display increased activity of senescence-associated β-galactosidase (SA-β-Gal, blue) and upregulated CAF marker α-SMA (red). (**B**). The expression of α-SMA, Vimentin, p53, p21 and p16 in fibroblasts were analyzed by Western blotting. Diagrams represent α-SMA, Vimentin, p53, p21 and p16 protein levels as means ± SEM of densitometric quantifications of two independent experiments (n = 2) normalized to GAPDH. Statistical analysis; *, *p* < 0.05. (**C**). Fibroblasts were subjected to a collagen contraction assay and imaged after 24 h. Quantification of the collagen contractile activity, statistical analysis, *, *p* < 0.05.

**Figure 2 genes-13-01579-f002:**
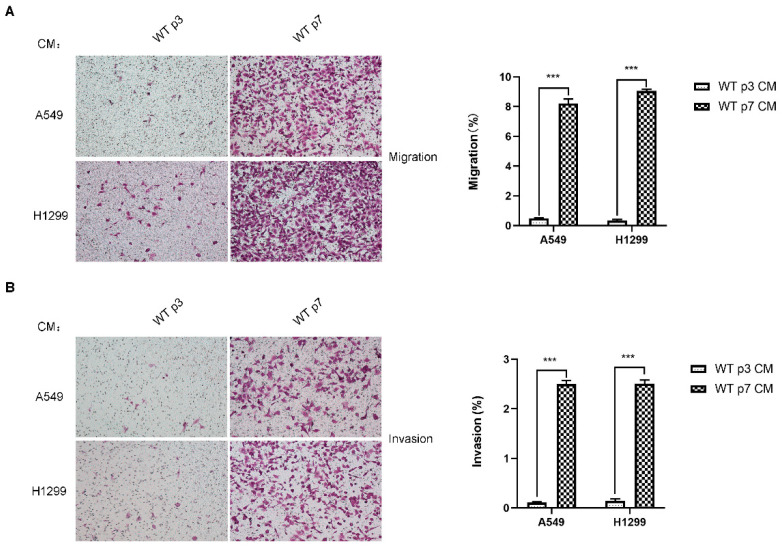
**Senescent fibroblasts promote the migration and invasion of lung cancer cells, H1299 and A549.** (**A**). Transwell migration assay of H1299 and A549 cells following co-culture with CM from young (p3) and senescent (p7) fibroblasts after 18 h. (**B**). Transwell invasion assay of H1299 and A549 cells following co-culture with CM from young (p3) and senescent (p7) fibroblasts after 18 h. Average migration/invasion ±SEM from three independent experiments performed. Statistical analysis; ***, *p* < 0.001.

**Figure 3 genes-13-01579-f003:**
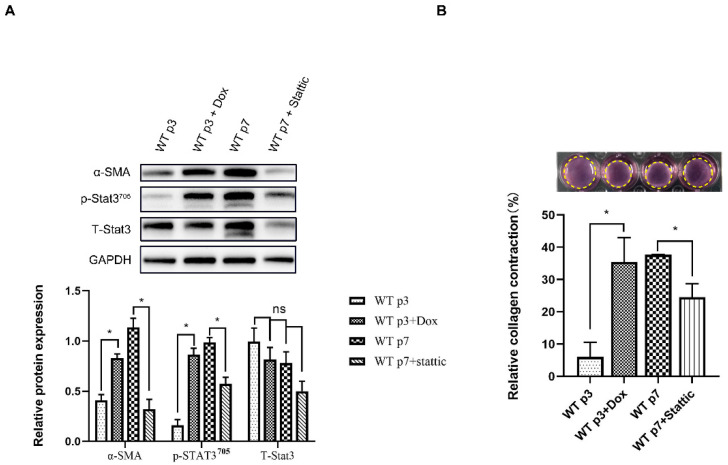
**Senescent fibroblasts****display CAF Properties through the Stat3 pathway.** (**A**). Western blotting analysis for α-SMA, p-Stat3 and Stat3 of young fibroblasts (p3), young fibroblasts (treated with dox or replicative senescence) and senescent fibroblasts treated by Stattic. Quantification of the Western blotting and statistical analysis; *, *p* < 0.05. and ns indicate non-significance. (**B**). Fibroblasts as in A were subjected to a collagen contraction assay.

**Figure 4 genes-13-01579-f004:**
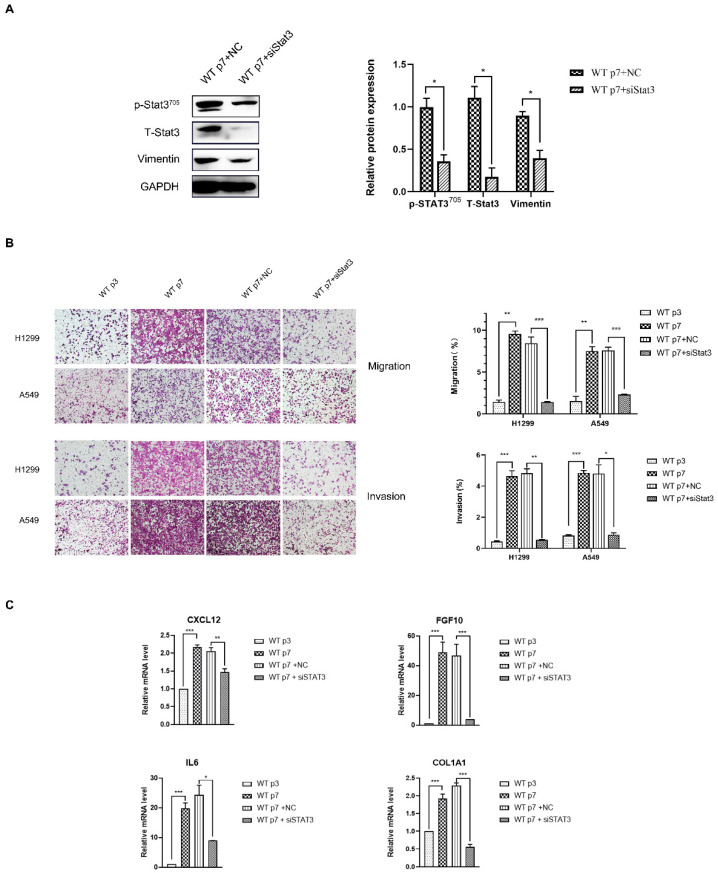
**Senescent fibroblasts****promote the migration and invasion****of lung cancer cells H1299 and A549****through the Stat3 pathway.** (**A**). Western blotting analysis of Stat3 and Vimentin expression in senescent fibroblasts and Stat3-transfected senescent fibroblasts. Quantification of the Western blotting and statistical analysis. *, *p* < 0.05. (**B**). Transwell assays were performed to evaluate the migration and invasion of H1299 and A549, following a co-culture with fibroblasts. An average migration/invasion ±SEM from three independent experiments was performed. Statistical analysis. **, *p* < 0.01, ***, *p* < 0.001. (**C**). Quantification of CXCL12, FGF10, IL6 and COL1A1 mRNA from fibroblasts by qRT-PCR. Values (mean ± SEM) were derived from three independent experiments. *, *p* < 0.05 **, *p* < 0.01, ***, *p* < 0.001.

**Figure 5 genes-13-01579-f005:**
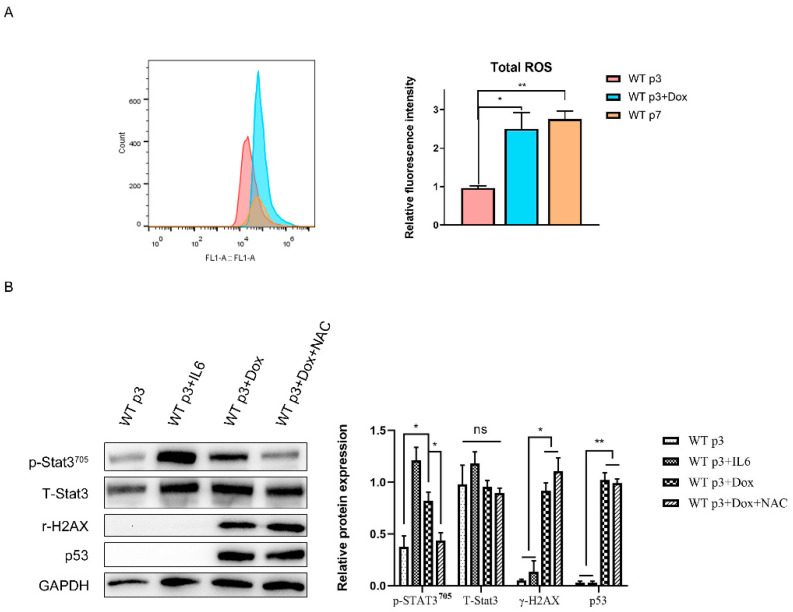
**High level ROS active Stat3 pathway****in senescent fibroblasts****.** (**A**). The ROS level of young fibroblasts (p3), senescent fibroblasts (treated with dox or replicative senescence). Quantification of relative ROS level with relative fluorescence intensity. Statistical analysis. *, *p* < 0.05, **, *p* < 0.01. (**B**). The expression of Stat3 and p-Stat3, r-H12x and p53 in fibroblasts were analyzed by Western blotting. They were treated with IL6 as a positive control. Statistical analysis; *, *p* < 0.05, **, *p* < 0.01 and ns indicate non-significance.

## Data Availability

Not applicable.

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
