# Peer review of "Senescent Fibroblasts Generate a CAF Phenotype through the Stat3 Pathway"

_genes, 2022, doi:10.3390/genes13091579_

Round 1
Reviewer 1 Report
The manuscript by Li et al describes the regulation of CAF-like phenotype in senescent fibroblasts through the activation of Stat3 pathway via intracellular ROS accumulation. The phenomenon has been previously described by Yasuda et al in gastric cancer (2021, Cell reports), however, the findings are novel in terms of lung cancer.
Following revisions are required to conclusively agree on the findings of their manuscript:
1A Quantification of SA beta gal and alpha-sma
1B Quantify western blot
1C. For the collagen contractile activity, clearer images showing visible differences are required. It is not possible to visually conclude a difference in the images provided. Quantification of collagen contractile activity is also required.
2A, B. Quantification of cell migration and invasion is required.
3A and B should be made into separate figures. Quantification of both western blot and collagen contractile activity is required.
3C. Quantification of cell migration and invasion is required.
3D. Provided larger, sharper images of the bar graph. State any statistical difference in gene expression by denoting significance.
4A. Provide larger, sharper histogram for flow cytometry. Was this experiment performed in triplicates? What was the negative control used for gating?
Quantify the results and denote statistical significance.
Author Response
Dear Reviewer,
First, we would like to thank you for the thoughtful comments and suggestions. We would also like to express our gratitude for providing us with an opportunity to revise our manuscript titled “ Senescent fibroblast generates a CAF phenotype though Stat3 pathway” (ID: 1838982 ). Those comments are valuable and very helpful. We have read through the comments carefully and have made corrections. Based on the instructions provided in your letter, we uploaded the file of the revised manuscript. Red represents modification. A point-by-point response to the comments is attached below.
1A Quantification of SA beta gal and alpha-sma
Answer:
“The late-passage(p=7)MEF exhibited a classical morphology characterized by increased granularity, enlarged and flattened shape. In addition, We confirmed senescence by the increase in fibroblasts staining positive for senescence-associated β-gal (SA-β-gal, Figure 1A, 17.9±0.2%).”were added in the paper.
“97% senescent fibroblasts with α-SMA-positive staining(Figure 1A)” were added in the paper.
1B Quantify western blot
Answer:
Quantification of the western blot were added in the Figure 1B.
1C. For the collagen contractile activity, clearer images showing visible differences are required. It is not possible to visually conclude a difference in the images provided. Quantification of collagen contractile activity is also required.
Answer:
Quantification of collagen contractile activity was added in the Figure 1C.
2A, B. Quantification of cell migration and invasion is required.
Answer:
Quantification of cell migration and invasion were added in the Figure 2A,2B.
3A and B should be made into separate figures. Quantification of both western blot and collagen contractile activity is required.
Answer:
Western blot and collagen contractile activity were made into figure 3A and 3B, and quantification of both western blot and collagen contractile activity were added in the figure.
3C. Quantification of cell migration and invasion is required.
Answer:
Quantification of cell migration and invasion were added in the Figure 3C
3D. Provided larger, sharper images of the bar graph. State any statistical difference in gene expression by denoting significance.
Answer:
We provided the new images of the bar graph and statistical analysis were added in the Figure 3D. *,p<0.05 **,p<0.01,***,p<0.001
4A. Provide larger, sharper histogram for flow cytometry. Was this experiment performed in triplicates? What was the negative control used for gating?
Quantify the results and denote statistical significance.
Answer:
We did the experiment in triplicates.
We used the fibroblast that did not stain by ROS reagent as negative control.
The the mean florescence intensity of DCF was used to quantify. The quantification of the results and statistical analysis were added in the Figure 4A. **,p<0.01.
The quantification of western blot were added in the figure 4B.
We really appreciate your efforts in reviewing our manuscript. We believe that the revised version of the manuscript represents a significant improvement, and your favorable consideration would be greatly appreciated.
Sincerely,
Xiaoming Wu

Reviewer 2 Report
1. Please revise title. Abbreviations such as "CAF-like" make it difficult to understand what your research encompasses.
2. Please review grammar of title and of entire manuscript. Extensive language editing of M/S is required.
3. The involvement of the JAK-STAT pathway has already been described
-
Albrengues, J. et al. Epigenetic switch drives the conversion of fibroblasts into proinvasive cancer-associated fibroblasts. Nat. Commun. 6, 10204 (2015).rn to ref 59 in article
-
Albrengues, J. et al. LIF mediates proinvasive activation of stromal fibroblasts in cancer. Cell Rep. 7, 1664–1678 (2014). (amougst many others)
Please advise how your report adds to the general knowledge of this topic.
4. In 2.1 "As described previously, we harvested MEF with WT genotype: Please give the reference in which this is described. It is not clear how the fibroblasts were quantified and identified after isolation to confirm cell type/phenotype.
5. Methods are not described in sufficient detail to replicate. Also, manufacturer information of reagents/kits not adequately done.
6. All abbreviations must be defined on first use.
7. Please include primer design metrics in suppl data
8. 2.6 "(WT p3, WT p7) or WT p3, WT p7, WT p7+NC, WT p7+siStat3" These are not adequately defined.
9. 2.6 - a grid pattern for taking micrographs would be more rigorous than random fields. How many replicates per repeat?
10. Methods do not seem to include controls - these are needed in order to contrast results (positive method control and negative control to demonstrate validity of method and contrast results). When getting to the results, they may have been included in some experiments - but this must be clear in the methods.
11. Results are incredibly difficult to follow. This may be a combination of the poor language which needs editing, a lack in the method description and defining of all parameters, as well as insufficient background and motivation for this study in the introduction. One gets the feeling that the lung cancer cells were added in as an afterthought. The logic and flow of the experimental design is difficult to follow, and the results need to be discussed in more detail, with values and stats. The figures are at times not clear – eg 3A and 3D, 4A
12. Fig 1 A. Very few cells appear to be stained green with SA-beta-Gal (by far the minority). This brings into question the other results - how sure is one that these are indeed senescent fibroblasts being investigated?
13 "In our study, Co-IP demonstrated that wt-p53 inhitited SHP2 binding with Stat3 in 293T(not shown in the paper)." why is it not shown? Where is it?
14. "We also further propose a novel Stat3 dependent targetable mechanism that is instrumental in mediating lung cancer cells migration and invasion." This is not elucidated well in the discussion.
15. The discussion is superficial and very rudimentary.
16. Errors in referencing format
17. This manuscript would need major restructuring and editing to be reassessed for consideration. I do understand that it must be enormously difficult to write up your science in a language that is not your own, but sadly so much of what you are doing is lost in translation at the moment.
Author Response
Dear Reviewer,
First, we would like to thank you for the thoughtful comments and suggestions. We would also like to express our gratitude for providing us with an opportunity to revise our manuscript titled “ Senescent fibroblast generates a CAF phenotype though Stat3 pathway” (ID: 1838982 ). Those comments are valuable and very helpful. We have read through the comments carefully and have made corrections. Based on the instructions provided in your letter, we uploaded the file of the revised manuscript. Red represents modification. A point-by-point response to the comments is attached below.
- Please revise title. Abbreviations such as "CAF-like" make it difficult to understand what your research encompasses.
Reply:
We replaced "CAF-like" with “CAF” in the title.
- Please review grammar of title and of entire manuscript. Extensive language editing of M/S is required.
Reply:
We replaced "CAF-like" with “CAF” in the paper. And we have edited the manuscript again.
- The involvement of the JAK-STAT pathway has already been described. Please advise how your report adds to the general knowledge of this topic.
Reply:
JAK-STAT pathway activited CAFs in normal fibroblasts has already been described. However, it is unknown whether Stat3 pathway in senescent fibroblasts could play the same role. How senescent fibroblast promote tumorigenesis is also unknown. We discribed the JAK-STAT pathway that linked CAFs activation with fibroblast senescence. JAK-STAT pathway confers tumor-promoting activity on senescent fibroblasts.
- In 2.1 "As described previously, we harvested MEF with WT genotype: Please give the reference in which this is described. It is not clear how the fibroblasts were quantified and identified after isolation to confirm cell type/phenotype.
Reply: “MEF were harvested and prepared from individual day 13.5 embryos obtained from wild-type C57/B6 mice.” was revised in the paper. MEF were prepared according to standard protocols(Zindy, F., et al., Expression of the p16INK4a tumor suppressor versus other INK4 family members during mouse development and aging. Oncogene, 1997. 15(2): p. 203-11.)
- Methods are not described in sufficient detail to replicate. Also, manufacturer information of reagents/kits not adequately done.
Reply: We added experimental details and reference in the Methods. The“manufacturer information of reagents/kits” were added in the paper. “Briefly, fibroblasts were seeded in 10-cm dish, grown to 70-80% confluence, and then transfected with 20 nM siRNA using lipofectamine 3000 (Invitrogen, Carlsbad, United States). 48hs later, cells were collected for western-blot analyses to determine the knockdown efficiency.” were added in the Method of 2.2.
- All abbreviations must be defined on first use.
Reply: All abbreviations have been defined on first use.
- Please include primer design metrics in suppl data
Reply: “The sequences of each forward (F) and reverse (R) primer used for qRT-PCR were as follows:
CXCL12-F: 5'-TGCATCAGTGACGGTAAACCA-3',
CXCL12-R:5'-CACAGTTTGGAGTGTTGAGGAT-3;
IL6-F: 5'-TAGTCCTTCCTACCCCAATTTCC-3',
IL6-R: 5'-TTGGTCCTTAGCCACTCCTTC-3';
FGF10-F: 5ʹ-CACATTGTGCCTCAGCCTTTC-3ʹ,
FGF10-R: 5ʹ-CTCTATTCTCTCTTTCAGCTTAC-3ʹ;
Col1A1-F: 5ʹ-CATTGTGTATGCAGCTGACTTC-3ʹ,
Col1A1-R:5ʹ-CGCAAAGAGTCTACATGTCTAGG-3ʹ;
β-actin-F: 5ʹAGAGGGAAATCGTGCGTGAC−3ʹ,
β-actin-R: 5ʹ-CAATAGTGATG ACCTGGCCGT−3ʹ; ” were added in the paper.
- 2.6 "(WT p3, WT p7) or WT p3, WT p7, WT p7+NC, WT p7+siStat3" These are not adequately defined.
Reply: “young fibroblasts: WT p3ï¼›senescent fibroblasts: WT p7ï¼›WT p7+NC:In siRNA transfection,andom sequences was used as negative control(NC).” were added in the paper.
- 2.6 - a grid pattern for taking micrographs would be more rigorous than random fields. How many replicates per repeat?
Reply: Each experiment was repeated three times. Each sample was did in triplicate
- Methods do not seem to include controls - these are needed in order to contrast results (positive method control and negative control to demonstrate validity of method and contrast results). When getting to the results, they may have been included in some experiments - but this must be clear in the methods.
Reply:
“Fibroblasts treated with H2O2 were used as positive control, and unstained fibroblasts were used as negative control.” were added in the Method of ROS.
“WT p3 treated with IL6 was used as positive control.” were added in the Method of western blot and figure 4B.
11.Results are incredibly difficult to follow. This may be a combination of the poor language which needs editing, a lack in the method description and defining of all parameters, as well as insufficient background and motivation for this study in the introduction. One gets the feeling that the lung cancer cells were added in as an afterthought. The logic and flow of the experimental design is difficult to follow, and the results need to be discussed in more detail, with values and stats. The figures are at times not clear – eg 3A and 3D, 4A
Reply:
We quantified the all of western blot, collagen contractile activity, cell migration and invasion and ROS.
“Lung cancer is the leading cause of cancer mortality worldwide[19]. In recent years, age has been recognized to play a important role in the initiation and progres-sion of lung cancer[20, 21]. Senescent fibroblasts associated with lung cancer progres-sion has attracted the attention of researchers.” were added in the introduction.
- Fig 1 A. Very few cells appear to be stained green with SA-beta-Gal (by far the minority). This brings into question the other results - how sure is one that these are indeed senescent fibroblasts being investigated?
Reply:
“The senescent MEF exhibited a classical senescence-associated morphology characterized by increased granularity, enlarged and flattened shape.” In addition, We confirmed senescence by the increase in fibroblasts staining positive for senescence-associated β-gal.
13 "In our study, Co-IP demonstrated that wt-p53 inhitited SHP2 binding with Stat3 in 293T(not shown in the paper)." why is it not shown? Where is it?
Reply: We found wt-p53 inhitited SHP2 binding with Stat3 in 293T. However, we did not get the same results in the senescent fibroblasts. We need longer time to perform the experiment. So we did not add thess results in the paper.
- "We also further propose a novel Stat3 dependent targetable mechanism that is instrumental in mediating lung cancer cells migration and invasion." This is not elucidated well in the discussion.
Reply:
“However,blockade of Stat3 pathway may result in an impairment of tumor immune surveillance[36] and inhibition of Stat3 pathway did not affect normal cell survival[37].These different Stat3 pathway functions in tumorigenesis need to be taken into account in future clinical trials targeting the Stat3 pathway.” were added in the discussion.
- The discussion is superficial and very rudimentary.
Reply:
We revised the discussion.
- Errors in referencing format
Reply: referencing format was revised in the paper.
- This manuscript would need major restructuring and editing to be reassessed for consideration. I do understand that it must be enormously difficult to write up your science in a language that is not your own, but sadly so much of what you are doing is lost in translation at the moment.
Reply:
We really appreciate your efforts in reviewing our manuscript. We believe that the revised version of the manuscript represents a significant improvement, and your favorable consideration would be greatly appreciated.
Best regards!
Yours sincerely,
Xiaoming Wu

Round 2
Reviewer 1 Report
The changes made are sufficient to accept the manuscript in the current form.
Author Response
Dear Reviewer,
Thank you for the valuable comments. The manuscript has been sent to MDPI for English editing, English editing ID: English-49600.
Yours sincerely,
Xiaoming Wu
Reviewer 2 Report
The manuscript entitled Senescent fibroblast generates a CAF phenotype though Stat3 pathway” has been resubmitted after the first review process. Please find comments on the revision:
1. The title still contains acronyms not well recognised, and is not yet grammatically correct.
2. Grammar, punctuation, formatting and syntax have not been sufficiently addressed. I do suggest submission to an English language editing service, which I am sure this journal offers.
3. The abstract does not convey the distilled message acuratly, and does not make it clear that this study is in lung cancer cells which makes it different from previous studies.
4. It is not usual to discuss findings of the current study in the introduction. Furthermore, the introduction remains disjointed (particularly the last 1/3rd) and scant of background needed to substantiate the study.
5. Manufacturer references must be standardised throughout eg (company, city, state, country). Either all must have catalogue numbers, or none. Abbreviations must be consistent, and once defined, only use the abbreviation.
6. Not all abbreviations have been defined. There is inconsistency in abbreviation use.
7. “2.1. Mice, Cell and treatment”. These types of headings are not scientifically or grammatically acceptable.
8. I would not be able to replicate the methods described here. The information added is insufficient and not accurate enough.
9. Results are marginally improved, but quite randomly statement without a smooth and comprehensive progression.
10. Fig 1: There is only a very mild increase in SA-beta-Gal staining in Fig 1A. This is not really convincing, and one would deduce that only a very small portion of those fibroblasts are senescent. This has not really been adequately addressed. Why is there a discrepancy between the staining and the protein expression? WB are still not analysed statistically to ascertain significance of fold-increase, and it is difficult to see what is represented in 1C. All the RAW WB images ae required please. Legend is incomplete and poorly phrased, with inconsistent use of abbreviations.
11. Findings are not adequately discussed in the text. It remains very difficult to follow, and vague reports of results are not scientifically accurate. Language and syntax errors have not been corrected.
12. Point 13. These experiments must be concluded and added to the M/S or not mentioned at all. Particularly if results are discordant.
13. Point 9 has not been adequately addressed.
14. Figure 3: figures are too small. One cannot make out the detail on the various figures. See previous comments on WB.
15. “From these experiments we concluded that high level ROS modulate CAF functions though Stat3 pathway”. This is quite a sweeping statement. How sure are you that no other pathway is involved? Or other signals? Rephrase to be less reaching. This comment may be applied to mnay of the reaching conclusions, such as “We also present evidence to indicate that that senescent fibroblasts and Lung cancer progression are closely linked.”
16. All the same concerns with Fig 4.
17. The discussion is marginally improved, but not yet sufficient. No caveats, delineations or experimental shortfalls were discussed at all. It remains superficial and rudimentary.
18. The conclusion could be more encompassing, and more directed future studies may be of more interest than a sweeping non-specific statement.
19. Please provide ethics approval number from a recognised body for this study.
Author Response
Dear Reviewer,
First, we would like to thank you for the thoughtful comments and suggestions. We have read through the comments carefully and have made corrections. Based on the instructions provided in your letter, we uploaded the file of the revised manuscript. Red represents modification. A point-by-point response to the comments is attached below.
